# Experiences of tobacco cessation including a prescription approach among patients in Swedish primary health care with a focus on socioeconomically disadvantaged areas

**Anne Leppänen**[1]*, **Solvig Ekblad**[1,2], **Tanja Tomson**[1]

1 Department of Learning, Informatics, Management and Ethics, Karolinska Institutet, Stockholm, Sweden,
2 Academic Primary Health Care Center, Region Stockholm, Stockholm, Sweden

* anne.leppanen@ki.se

**Data Availability Statement:** Transcribed interviews (in Swedish) are stored at Karolinska Institutet, Department on Learning, Informatics,

## Abstract

### Background

Tobacco Cessation on Prescription (TCP) is a new intervention that is being evaluated in socioeconomically disadvantaged areas in Swedish primary health care (PHC). Patients' perceptions of TCP are important to understand as this may have implications for the acceptability and adherence to treatment and explain cessation outcomes. Patients' general experiences of tobacco cessation are also important to explore to improve cessation support in this setting.

### Aim

To explore experiences of tobacco cessation and TCP among patients in Swedish PHC focusing on socioeconomically disadvantaged areas.

### Methods

Inductive content analysis of transcripts from eight semi-structured interviews with patients recruited from the intervention group in a randomized controlled trial evaluating the effectiveness of TCP in socioeconomically disadvantaged areas in PHC in Stockholm.

### Results

Two themes were identified: *needing individualized support to quit*, *taking differences in patients' experiences of tobacco use and cessation into account*, acknowledging individual factors such as impact of health and wellbeing on tobacco use and differing attitudes towards tobacco and cessation and *needing a supportive environment to facilitate tobacco cessation*, taking contextual factors like professional support from the health care system, the importance of the social environment and supportive societal structures into account. Regarding TCP, the prescription form was perceived as useful for providers but did not appear to have a direct impact on tobacco cessation from the informants' perspective.

Management and Ethics. The data set cannot be shared publicly due to ethical and legal restrictions since it contains potentially identifying and sensitive information that compromises the informants' confidentiality. This would violate the ethical approval obtained from the Regional Ethical Review Board in Stockholm [ref: 2015/207-31] as well as the data protection policy of Karolinska Institutet. Data are available on request from the Registrar's Office at Karolinska Institutet (contact via registrator@ki.se) for researchers who meet the criteria for access to confidential data.

**Funding:** This study was funded by the Public Health Agency of Sweden [grant no: 0217-2017-6.2]. The funder had no role in study design, data collection and analysis, decision to publish, or preparation of the manuscript.

**Competing interests:** The authors have declared that no competing interests exist.

However, individualized counseling from a tobacco cessation specialist, an empathetic approach in the treatment and long-term follow-up was considered important.

## Conclusion

A holistic approach may be needed in cessation treatment, combined with interventions outside the health care system, to facilitate tobacco cessation among patients in socioeconomically disadvantaged areas in Swedish PHC. The TCP prescription form may be helpful for PHC providers but counseling and follow-up appear to be the most important components of TCP for patients in this setting.

## Introduction

Smoking has been identified as one of the leading preventable risk factors for ill-health in the world [1]. In Sweden, the prevalence of daily smoking is 7% [2]. Furthermore, 18% of men and 4% of women use the smokeless tobacco product "snus" on a daily basis [2]. Although the prevalence of daily smoking is relatively low in Sweden compared to other countries [3], it is almost three times higher among those with the lowest compared to those with the highest socioeconomic status [2]. In addition to this unequal distribution between different social groups, socioeconomically disadvantaged groups (here defined as groups with a lower socioeconomic status, e.g. lower income, educational level or occupational status) are more negatively affected by their tobacco use and experience greater difficulties in quitting compared to their counterparts [4, 5]. During the last years, several policy goals to reduce the prevalence of tobacco use and related inequalities in health have been adopted in Sweden [6, 7]. The national treatment guidelines for tobacco cessation describe which treatments that are effective and recommend that treatment to socioeconomically disadvantaged groups should be prioritized but they do not provide any recommendations for how such services should be organized [8].

Primary health care (PHC) has the main responsibility for health promotion and disease prevention in the Swedish health care system [9] and socioeconomically disadvantaged groups visit PHC more often than their counterparts [10]. Therefore, PHC can be seen as an important stakeholder in providing cessation support to lower socioeconomic groups. Studies from the UK National Health Service show that intensive treatments for smoking cessation are effective in helping smokers to quit [11] and successful in reaching smokers from disadvantaged communities [12]. However, relatively few smokers that visit PHC in Sweden are identified and offered support to quit [13]. Organizational barriers such as high workload and lack of staff, time and resources to work with health promotion have been found to present challenges for PHC to prioritize such activities [14, 15].

Tobacco Cessation on Prescription (TCP) is a new PHC intervention that could facilitate tobacco cessation in Swedish PHC [16]. It is inspired by Physical Activity on Prescription (PAP), which is a physical activity referral scheme that has been found effective in increasing physical activity and improving health and quality of life in the general population in Sweden since 2001 [17]. TCP has been developed based on PAP, on the findings from a qualitative study that explored the perceived feasibility and optimal design of TCP [16] and on clinical guidelines for tobacco cessation in Sweden [18]. It consists of counseling for at least 10 minutes, an individualized prescription of tobacco cessation treatment with options for further counseling (referral to a health care provider with more competence or the Swedish National Tobacco Quitline), pharmacotherapy (nicotine replacement therapy, varenicline, bupropion),

other measures for cessation (physical activity, strategies to cope with withdrawal symptoms), support for self-management (questions for self-reflection, reference to mobile applications and websites) and follow-up (by telephone or revisit) on at least one occasion [19].

The effectiveness of TCP is currently being evaluated in a randomized controlled trial (RCT) in the Swedish PHC setting with a focus on socioeconomically disadvantaged areas [19]. In addition, a qualitative study has recently explored PHC providers' perceived barriers and facilitators of implementing TCP [20]. The study found that TCP was perceived positively by PHC providers but that socioeconomically disadvantaged groups may face specific barriers to quit their tobacco use. Perceived low motivation to quit in this target group was explained by high exposure to tobacco in the social environment, a negative attitude towards treatment and tobacco being used as a coping strategy to deal with life stress. Furthermore, access to treatment for this target group was perceived to be limited by costs of treatment, long waiting times and barriers to attend physical visits for counseling. This was partly explained by how tobacco cessation treatment was organized in PHC with a perceived lack of time, resources and capacity among PHC providers to work with tobacco cessation. Several studies have been conducted on TCP in PHC in Stockholm but it has not been formally implemented in this setting. To inform a potential implementation of TCP, it is important to complement data on effectiveness and PHC providers' experiences with patients' own perceptions of TCP as this may have implications for the acceptability and adherence to treatment and may help to explain cessation outcomes. The relationship between socioeconomic status and tobacco cessation has previously been investigated in the Swedish context [21] but there is a knowledge gap regarding experiences of tobacco cessation among patients in lower socioeconomic settings in Swedish PHC. These experiences may to some extent be context-specific and thus important to explore in order to improve cessation support in this particular setting. Therefore, the aim of this study was to explore experiences of tobacco cessation and TCP among patients in Swedish PHC focusing on socioeconomically disadvantaged areas.

## Methods

### Study design

The study applied an exploratory qualitative study design based on inductive content analysis [22, 23] of transcripts from semi-structured interviews with patients who had personal experience of tobacco cessation and TCP in the PHC setting in Stockholm.

### Study informants and setting

Patients with personal experience of tobacco cessation and TCP with willingness to participate in the current study were purposefully sampled from the intervention group in an ongoing RCT, evaluating the effectiveness of TCP in 18 PHC centers located in areas with a higher proportion of socioeconomic disadvantage in Region Stockholm. Further details on the RCT and its implementation are provided in the trial registration [ISRCTN 11498135] and the published study protocol [19]. Purposeful sampling is recommended in qualitative studies to collect data from those with the best knowledge of the phenomenon under study [24]. Since TCP has not been formally implemented in clinical practice, the patients from the RCT were the only patients with experience of this intervention. These patients were also expected to have some experience of trying to quit their tobacco use from their participation in the RCT. Thus, they were considered the most suitable informants for the current study. Patients that had responded to the last follow-up questionnaire in the RCT in May 2018 to January 2019 were sent an information letter about the study via mail. The patients were contacted within six months after completing their participation in the RCT to avoid any influence of the

interviews on the RCT results and to minimize the risk of recall bias. The letter was sent by one of the researchers (AL) and followed up with up to five telephone calls, one voice mail and one text message two to four weeks after the information letter was sent to invite the patients to participate. The informants were offered a 100 SEK (approximately 10 USD) gift certificate as an incentive to participate in the study.

Recruitment of informants in this study continued as more patients responded to the last follow-up questionnaire in the RCT until the information power produced was considered sufficient [25]. Saturation is a term that is closely linked to a specific methodology, while information power is an alternative concept that can be used in qualitative research to determine the number of participants needed in a study [25]. This concept has been developed based on available literature about the current state of the art regarding sample size in qualitative studies and weaknesses in these standards [25]. Information power is relative and informed by the specificity of the aim and sample of a study, the quality of the dialogue and whether established theory has been applied [25]. Due to the inductive nature of the current study, it was not relevant to apply any theory to guide the data analysis. However, the aim and the specificity of the sample in this study was considered narrow and the quality of the dialogue strong. Therefore, a small sample size was considered appropriate.

In total, 23 patients from five of the intervention PHC centers in the RCT were contacted. Of these, eight patients from three PHC centers agreed to participate, five patients declined and the other ten patients did not respond to the follow-up calls or text messages. Reported reasons for non-participation were lack of interest and time to participate in the study. Informant characteristics are presented in Table 1. Non-participants had similar characteristics but were somewhat younger and more often male.

## Data collection

Data was collected in December 2018 and January 2019 through semi-structured interviews in conversational form. The informants were asked about their experiences of tobacco cessation and TCP in the PHC setting, for example: "How did you perceive the support you received from your PHC center to quit your tobacco use?" and "How did you perceive the form [TCP]?" The interview guide was pilot tested during the first interview that was included in the data collection. As a result, one broad open-ended question about general experiences of tobacco cessation was added to the interview guide and one long question was divided into two separate questions. The interview guide is presented in the S1 Appendix. A copy of the TCP prescription form was also provided when the informants were asked about TCP. The TCP prescription form is presented in the S2 Appendix.

Four interviews were conducted face-to-face in private rooms at public libraries close to the patients' homes, whereas four interviews were conducted via telephone based on the patients' preferences. The interviews were 21–87 (on average 42) minutes long, conducted in Swedish, audio-recorded and transcribed verbatim by AL who has formal training in qualitative research methods, previous experience from two qualitative studies and a background in public health science and health promotion. AL was the coordinator for the RCT and also involved in the development of TCP.

In total, the transcripts were 72 single-spaced A4 pages long and took approximately 30 hours to transcribe. Field notes summarizing the interview content and situation were also taken to facilitate the transcription of the interviews, to enhance the credibility of the findings and to reflect on strengths and limitations of the study but they were not included in the data analysis. Written informed consent was obtained from all informants before the start of each

**Table 1. Informant characteristics.**

| Characteristic | Informants n = 8 |
|---|---|
| Age (years)[a] | |
| Mean | 60.5 |
| Range | 28–77 |
| Gender[a], n (%) | |
| Male | 2 (25) |
| Female | 6 (75) |
| Socioeconomic disadvantage[a, b], n (%) | |
| Yes | 7 (87.5) |
| No | 1 (12.5) |
| Type of tobacco used[a], n (%) | |
| Cigarettes | 8 (100) |
| Other | 0 (0) |
| Heaviness of Smoking Index (nicotine dependence)[a], n (%) | |
| Low | 1 (12.5) |
| Medium | 4 (50) |
| High | 3 (37.5) |
| Fidelity to TCP components[a], n (%) | |
| Counseling ($\geq$10 minutes) | 8 (100) |
| Prescription form | 7 (87.5) |
| Follow-up ($\geq$1 occasion) | 6 (75) |
| Current tobacco use, n (%) | |
| Quit use | 3 (37.5) |
| Reduced use | 2 (25) |
| Continued use | 3 (37.5) |

[a] Data from RCT, collected up to 19 months before the interviews in the current study.

[b] Lower educational level ($\leq$12 years) or occupational status (not employed).

interview and ethical approval was obtained from the Regional Ethical Review Board in Stockholm [ref: 2015/207-31].

## Data analysis

As suggested by Krippendorff, both the manifest and latent content of the transcripts was considered in the analysis [26], meaning that both the explicit and implicit meaning of the informants' accounts was taken into consideration. First, AL, SE and TT independently read the transcripts several times to get an overview of the data. SE has extensive experience in teaching and conducting qualitative research on vulnerable groups in different cultural settings and a background in clinical psychology and multicultural health and care research. TT has experience from several qualitative research studies in different contexts globally and a background in nursing, prevention and public health science. TT also led the development of TCP. Passages in the text related to the aim of the study were extracted from the transcripts independently by AL and SE and brought together to constitute the unit of analysis. AL and SE independently identified meaning units (words, sentences or paragraphs containing aspects related to each other through their content and context), condensed them, abstracted them and labeled them as codes [22, 23]. Abstraction here refers to the description and interpretation of the meaning units on a higher logical level [22]. This was done inductively, meaning

that the analysis was data-driven as opposed to theory-driven [23]. Approximately one third of the codes were then compared through random checks to enhance credibility. Next, the codes were clustered into categories and sub-categories by AL. The categories and sub-categories were reviewed by SE and TT and discussed in meetings among all three authors until consensus was reached. Member checking was also conducted with two representatives of the target group to enhance the credibility of the analysis. No software was used to aid the data analysis and no changes were made to the coding as a result of the member checking. A summary of the themes, categories, sub-categories and examples of codes are presented in Table 2. Furthermore, one to two unidentified quotes from the informants are presented for each category in the results to support the analysis [27]. The interview guide, the quotes and the content in Table 2 was translated from Swedish to English by the researchers and reviewed by a native English speaker.

## Results

### Theme 1: Needing individualized support to quit, taking differences in patients' experiences of tobacco use and cessation into account

This theme covered three categories that focused on the impact of health and wellbeing on tobacco use, contradictory attitudes and experiences regarding tobacco use and differing attitudes and experiences of tobacco cessation and support to quit. The theme reflected that the informants had different attitudes toward tobacco use and cessation and that these attitudes were influenced by their individual experiences and life situations.

**Impact of health and wellbeing on tobacco use.** Knowledge about and attitudes towards health risks of tobacco use varied among the patients depending on their awareness of diseases attributed to tobacco use and their fear of being affected by these diseases. Informants that reported awareness of tobacco-related health risks and fear of falling ill stated that this was a motivation for them to quit their tobacco use:

"If I was told that I had a precursor to lung cancer, I'd sure as hell quit smoking at once. That's obvious. Or [if I had] another serious disease".

[Informant 2]

Health gains from quitting were also mentioned as a motivation to quit. At the same time, some patients perceived that it could be too late to quit if they became seriously ill. For example, two informants were being investigated for cancer and said that a cancer diagnosis would have a negative impact on their motivation to quit. Otherwise, patients with personal experience of health problems from their tobacco use appeared to be more motivated to quit. Reported health consequences of tobacco use included cough, phlegm, diseases worsened by tobacco use and negative results on lung tests. Several informants also reported that they had quit their tobacco use in connection to a health-related event, such as a surgery, pregnancy or when risking a foot amputation.

Stress related to the patient's life situation or other health problems was perceived as both a reason for using tobacco and as one of the most important barriers to quitting:

"What [. . .] I think is the reason [for] both why I smoke and why I can't quit is that I've had both a tough upbringing and a tough life [. . .] in general with a lot of loss, a lot of diseases."

[Informant 7]

**Table 2. Themes, categories, sub-categories and examples of codes.**

| Theme 1: Needing individualized support to quit, taking differences in patients' experiences of tobacco use and cessation into account | | |
|---|---|---|
| **Category** | **Sub-category** | **Examples of codes** |
| Impact of health and wellbeing on tobacco use | Knowledge and attitudes towards health risks of tobacco use | Knowledge about health risks of tobacco use, underestimation of harmful effects of tobacco use, risks of tobacco use motivation to quit, fear of tobacco-related disease, health gains of tobacco cessation, too late to quit due to serious illness |
| | Experienced consequences of tobacco use | Feeling unwell from tobacco use, less relevant to quit without experienced consequences of tobacco use, feedback on lung tests, quitting in relation to a health-related event |
| | Life stress and other health problems | Loneliness, loss, illness or death among friends and family, life situation, living situation, mental illness, gout, diabetes, disability, pain, overweight, unhealthy diet, sleeping problems, physical inactivity, wellbeing |
| Contradictory attitudes and experiences regarding tobacco use | Tobacco use as the individuals responsibility | Own responsibility to quit, up to patient and no one else to quit, quitting for one's own sake |
| | Tobacco use as an addiction manifested in different ways | Strong addiction to tobacco, tobacco use similar to other substance abuses, tobacco use as a habit, tobacco use as situation-specific, tobacco as a friend, tobacco use as a distraction, tobacco use as a coping strategy |
| | Predominantly negative feelings about tobacco use | Disgust with smell of smoke, disgust with taste of tobacco, tough to be a smoker, tired of addiction to tobacco, not sensible to smoke, shame, guilt, self-anger, self-blame, tasty to smoke |
| Differing attitudes and experiences of tobacco cessation and support to quit | Differing readiness to change | Thoughts about quitting, decreased tobacco use, decision to quit, motivation to quit, ambivalence, self-efficacy to quit, lack of courage and energy to quit |
| | Differing knowledge and attitudes toward cessation support | Lack of awareness of cessation treatment, awareness of cessation clinics at hospitals, knowledge about pharmacotherapy, positive attitude toward tobacco cessation specialist, skepticism toward group counseling, worry about adverse effects of pharmacotherapy, distrust in counseling, trust in pharmacotherapy, trust in e-cigarettes, need for support |
| | Different experiences of tobacco cessation | Previous quit attempts, positive experiences of decreasing/quitting tobacco use, experience of telephone counseling, positive experience of pharmacotherapy, experienced adverse effects of pharmacotherapy, negative experience of pharmacotherapy, experience of e-cigarettes, experience of self-help courses, registration of tobacco use, stepwise decrease of tobacco use, replacement strategies to quit, adherence to cessation treatment, experience of TCP prescription form, TCP prescription form helpful for providers, TCP prescription form not for patients |
| Theme 2: Needing a supportive environment to facilitate tobacco cessation | | |
| **Category** | **Sub-category** | **Examples of codes** |
| Professional cessation support from the health care system | Well-organized cessation support from the health care system | Cessation support from a physician, comprehensive counseling from a tobacco cessation specialist, information about the harmful effects of tobacco use, information about treatment options for tobacco cessation, motivational strategies, regular contact with a cessation specialist, long-term follow-up, collaboration between providers regarding tobacco cessation, cessation support from tertiary care, need for more tobacco cessation specialists |
| | Trust in the health care system | Lack of trust in the health care system, negative experiences of the health care system, low expectations on preventive measures from health care, different expectations between patients and providers, trust in the tobacco cessation specialist |
| | Positive experience of cessation support | Friendly providers, satisfaction with cessation support, support from health care crucial to cessation outcomes, patient involvement in the treatment, cessation support on the patients terms |
| High impact of social environment on tobacco use | Change in social norms regarding tobacco use | Tobacco use was previously more socially acceptable, glamorous to smoke in another time, one friend who smokes now compared to everyone before, negative perception of tobacco users in society |
| | Impact of tobacco use in the social environment | Tobacco use in the social environment, peer pressure to smoke, smoking as a social activity, sense of community with other smokers, quit attempts in the social environment motivation to quit |
| | Social support in quitting | Informing friends about quit attempts, encouragement and support from the social environment to quit, avoiding relapse to not let others down, quitting together with someone else |

(*Continued*)

**Table 2.** (Continued)

| | | |
|---|---|---|
| Supportive societal structures facilitating tobacco cessation | Availability of tobacco products and cessation support | Distance to nearest purchase location, access to cigarettes, isolation restricting access to tobacco, over-the-counter sales of nicotine replacement therapy, marketing of available cessation support, access to counseling |
| | Affordability of tobacco products and cessation support | Expense of cigarettes, tobacco use a waste of money, access to cheaper cigarettes, expense of pharmacotherapy, encouragement from subsidized pharmacotherapy |
| | Legislation promoting tobacco cessation | Inconvenient to go outside to smoke, might as well quit due to smoke-free air laws, soon not allowed to smoke anywhere, smoke-free air laws introduced to protect non-smokers, pictorial health warnings create disgust |

Several informants also reported that their tobacco use increased when they were stressed. Reported reasons for stress were related to loneliness, loss, illness or death among friends and family, the patients' living situation, mental illness, pain, physical disabilities, sleeping problems, overweight, unhealthy diet and physical inactivity. Thus, wellbeing was considered an important facilitator for successful tobacco cessation.

**Contradictory attitudes and experiences regarding tobacco use.** The informants viewed tobacco use as the individual's responsibility, saying that it was up to each person to decide for themselves whether or not they should quit. At the same time, tobacco use was described as a very strong addiction that was often compared to that of other substance abuses. The way in which this addiction was manifested differed greatly between individuals. Some of the informants described their tobacco use as a force of habit, while others described it as more situation-specific. Some patients also described tobacco as a friend, distraction or way of coping with stress and other negative feelings.

Furthermore, the patients reported mixed but predominantly negative feelings regarding their tobacco use. Some acknowledged that it could taste good but most of the informants were unhappy about their use. Negative feelings mentioned were disgust with the smell of smoke and taste of tobacco in some situations, inconvenience or dissatisfaction with being a smoker or being addicted to tobacco, as well as self-anger, self-blame, shame and guilt:

"I don't like the smell of smoke [. . .]. I have to wash my sheets and such all the time. I don't like it but oh, what can I say? It's awful. It's stupid [. . .] I promise, I hate this so much".

[Informant 4]

**Differing attitudes and experiences of tobacco cessation and support to quit.** Attitudes towards quitting varied among the informants. Some informants reported a high motivation and self-efficacy to quit while others did not. The informants also reported that they were in different stages in their tobacco cessation processes. Knowledge about available cessation support differed between the informants, as did attitudes and beliefs regarding the effectiveness of treatment:

"There are people that are determined that say"Yes, starting tomorrow I will quit smoking" and then they do it. Then there are those who might need some help. It depends on who the person is and [. . .] what they themselves want, [. . .] if they are so strong that they can manage to quit on their own or if they need some help with either some medication or counseling".

[Informant 7]

Patients that expressed a high awareness of treatment options, high perceived need of support and trust in treatment appeared to be more open to use pharmacotherapy and counseling in order to quit their tobacco use. Trust in pharmacotherapy was generally high among the informants, although some informants were worried about adverse effects and were skeptical towards replacing tobacco with other nicotine products. Trust in counseling was expressed by fewer informants compared to trust in pharmacotherapy. The informants preferred individual counseling to group counseling but highlighted that treatment preferences are highly individual and that others may have different preferences. Some patients also expressed interest in the use of electronic cigarettes.

In addition, the informants had different experiences of quitting from no previous quit attempts to many, using different methods with different outcomes and perceived levels of difficulty. Most informants reported personal experience of pharmacotherapy and counseling but use of electronic cigarettes and self-help materials were also reported. Some informants had managed to decrease or quit their tobacco use entirely while others had continued their use. Registration of tobacco use, followed by a stepwise decrease was considered helpful by many informants. Behavioral replacement strategies and pharmacotherapies were also considered useful by some informants but prescription drugs were often associated with adverse effects, particularly nausea.

When asked about the TCP prescription form, most of the patients could not remember if they had received it or not but they reported that they recognized the majority of its content. How the prescription form was perceived depended on how it was supposed to be used. Some patients reported a positive or neutral attitude towards the TCP prescription form while others had a more negative attitude, particularly if the purpose of the prescription form was unclear to them. For example, the form was perceived as unnecessary or inappropriate if it was used as a substitute for counseling or without follow-up, if it was used for patients to fill out on their own or on patients who wanted to quit their tobacco use without support. Most of the informants reported that they perceived the TCP prescription form as helpful for health care providers, e.g. to be used as a tool when exploring patients' tobacco use and planning, communicating, documenting and following up tobacco cessation treatment. In contrast, it was not necessarily considered useful for them as patients:

> "For me as a patient, I don't see it [the form] as positive at all. It might be a good basis for the counselor or [. . .] I can imagine that it could be good for her or him to sit and fill this out during the conversation for it to then be registered or for some statistics to be kept or [to check] "Have I done everything [. . .], talked about everything that we should have talked about?" so it probably fills a function but not for me as a patient."

> [Informant 3]

### Theme 2: Needing a supportive environment to facilitate tobacco cessation

Three categories were included in the second theme. These described more contextual factors like professional cessation support from the health care system, impact of the social environment on tobacco use and supportive societal structures facilitating tobacco cessation. Here, the informants reflected on the role of their environment in supporting them to quit their tobacco use.

**Professional cessation support from the health care system.**    The informants reported that professional cessation support from the health care system was an important resource for those who needed it. It was seen as positive if the support was well organized, for example with

referral from physicians to tobacco cessation specialists. The informants described that the support they had received from tobacco cessation specialists was more comprehensive than the support they had received from other health care providers. The specialist support included information about health risks of tobacco use and available treatment options, mapping of tobacco use and support through motivational strategies. It was considered beneficial if the contact with the cessation specialist was frequent and regular with long-term follow-up. Collaboration and communication between different health care providers, including those in tertiary care, was also perceived as beneficial since several providers then were aware of patients' quit attempts, followed up their tobacco use, stressed the importance of quitting and provided encouragement.

Trust in the health care system also appeared to be important, as low expectations and negative experiences from previous contacts with health care were reported to decrease patients' willingness to turn to health care for cessation support. Different expectations between patients and providers were reported as reasons for disappointment and loss of trust. Lack of follow-up and continuity in cessation treatment and other contacts with health care were given as examples that contributed to this. Trust in the tobacco cessation specialist was considered particularly meaningful. This was perceived to be facilitated by the tobacco cessation specialist's competence, experience, enthusiasm, commitment, encouragement and non-judgmental approach in regards to tobacco cessation treatment. Trust could also be facilitated through a prolonged contact with the same PHC provider:

> "In general it doesn't work like that at the PHC center that you're lucky enough to have the same contact person but I have had that [the same tobacco cessation specialist] the whole time [. . .] Otherwise it would probably have been more difficult if you would have gone to different people. Then you wouldn't have built that trust in the same way".

> [Informant 3]

At the same time, the informants reflected on the vulnerability of only having one tobacco cessation specialist at the PHC center, saying that absence and lack of a substitute could lead to an interruption in the treatment or even relapse. Therefore, the informants recommended to have more than one provider with this competence at each PHC center.

Most of the patients reported a positive experience of their contact with PHC. Staff was considered friendly and the informants were generally satisfied with the cessation support they had received even if some of the informants had discontinued their treatment and had not managed to quit. Many informants also reported that they felt involved in their treatment and that cessation support had been provided on their terms:

> "I believe that they [the providers at the PHC center] have listened to [. . .] what I have wanted and not pressured me to go somewhere or call anywhere or make contact a certain amount of times [. . .] but I have been allowed to go at my own pace and I have been allowed to decide for myself and I think that's good".

> [Informant 6]

Several patients reported that the support they had received from PHC had influenced their tobacco use and helped them to either reduce or quit their use. Still, some patients reported that the support from PHC had been insufficient and that long-term follow-up could be improved.

**High impact of social environment on tobacco use.** The informants perceived that tobacco use had become less socially acceptable and that others had a more negative attitude towards tobacco users now compared to before. This was reported as a motivation to quit but could also contribute to social isolation for those who were unsuccessful in quitting, as some informants reported that they mainly smoked at home and that they avoided smoking in public and in some social settings. Tobacco use in the social environment also affected the informants' tobacco use:

"Really, the best support is that none of my friends that I hang out with smoke".

[Informant 3]

Furthermore, the informants reported a sense of community among smokers. Tobacco use was described as a social activity and smoking initiation and relapse were often associated with exposure to tobacco through partners, friends and family members. At the same time, encouragement and support to quit from the social environment were perceived as facilitators for tobacco cessation, as was quitting attempts among friends and family and quitting together with someone else.

**Supportive societal structures facilitating tobacco cessation.** The availability of tobacco products was perceived to have an impact on patients´ tobacco use as some of the informants suggested prohibition of sales and isolation as interventions to protect them from exposure to tobacco. The informants also reported that their consumption decreased when availability of tobacco was restricted, for example if the distance to the nearest purchase location was remote. Similarly, high visibility and availability of cessation support through marketing and over-the-counter sales of nicotine replacement therapies, promoted their use.

Affordability of tobacco products was also perceived to have an impact as tobacco use was considered expensive and the cost was reported as a motivation for some patients to quit. Still, some of the informants reported that they were unaffected by this since they had access to cheaper cigarettes. Pharmacotherapies for tobacco cessation were also perceived as expensive but subsidies could encourage patients to use them:

"[It would make it easier] if there was a possibility to help us [patients] with aids [pharmacotherapy] at a lower price. To encourage [us to quit smoking] even more."

[Informant 4]

Moreover, the informants reflected on the impact of legislations such as comprehensive smoke-free air laws and pictorial health warnings on tobacco products, saying that these led to inconvenience and could increase their motivation to quit:

"I haven't really wanted to [quit smoking] [. . .] but I guess it's time now. Soon you won't be allowed to smoke anywhere so it's just as well [. . .] There is now a law that you can't smoke in outdoor dining areas and on platforms and such so [. . .] I might as well quit.

[Informant 5]

## Discussion

To the authors' knowledge, this is the first study to explore experiences of tobacco cessation and TCP among patients in Swedish PHC focusing on socioeconomically disadvantaged areas.

The first theme identified was *needing individualized support to quit, taking differences in patients' experiences of tobacco use and cessation into account*, acknowledging individual factors such as impact of health and wellbeing on tobacco use and differing attitudes towards tobacco and cessation. The second theme was *needing a supportive environment to facilitate tobacco cessation*, taking contextual factors such as professional support from the health care system, the importance of the social environment and supportive societal structures into account.

In the interviews, the informants mainly reflected on their general experiences of tobacco cessation. They described a paradigm shift from tobacco use being encouraged to becoming a socially unacceptable behavior. This could be explained by the introduction of comprehensive smoke-free air laws and other tobacco control policies during the last decades that have contributed to the de-normalization and decreased prevalence of tobacco use [28]. The informants reflected on the effects that these policies had on them, suggesting that they could be beneficial. At the same time, they could stigmatize tobacco use [29] and create feelings of shame and guilt among tobacco users [30, 31], as reported in this study and elsewhere. This could be reinforced by the informants' view of tobacco use as the individual's responsibility, a view that has also been reported in previous research [32]. Still, tobacco use was often described in a negative way and as an addiction, indicating that it was an unwanted behavior among the informants. Previous studies have found that stigma can increase tobacco user's intention to quit [30, 33] but also decrease self-efficacy [34] and further marginalize and isolate those who are unable to quit [30, 31, 35]. Similar experiences were reported by the informants in this study.

Moreover, fear of stigma can be a barrier for patients to seek help to quit [36, 37] and disclose their tobacco use [37, 38] and failed quit attempts [28] to health care providers. It can also result in resistance [34]. This places high demands on an empathetic approach among health care providers when working with tobacco cessation. The informants in this study stated that trust and a good relationship between the patient and the tobacco cessation specialist was of particular importance. This has also been highlighted by PHC providers in this setting [20]. The importance of individualized cessation support also became apparent as each of the informants reported a unique life situation that affected their ability and readiness to quit. This is supported by previous research on tobacco cessation among socioeconomically disadvantaged groups, showing that most of the barriers to quit relate to individual circumstances [5]. Earlier studies show that personalized and non-judgmental services may encourage deprived smokers to quit [39]. Tailored cessation services and combined counseling and pharmacotherapy may be particularly effective in addressing high nicotine dependence and life stress in socioeconomically disadvantaged groups to support them in quitting [40, 41]. This is well-aligned with the intervention components included in TCP and the content on the TCP prescription form.

A previous study by the authors suggests that PHC providers have a good understanding of patients' barriers to tobacco cessation in the given setting [20]. Most of the informants in the current study were also satisfied with the cessation support that they had received from PHC. Most of the informants reported that they had decreased or quit their tobacco use and that the support from PHC had been crucial to their success. This indicates that health care can have an important role in tobacco cessation and in offering support to those who need it. Previous studies have found that cessation support from the health care system can help smokers to quit [11] but poorer outcomes among socioeconomically disadvantaged smokers may need to be compensated by investing more resources to reach this target group [40]. At the same time, the informants that had not managed to quit reported their health status, life situation, exposure to tobacco use in the social environment and high availability of tobacco products as the most important barriers to quitting. This suggests that cessation treatment alone is not enough

and that a more holistic approach may be needed, both in cessation treatment and in combining cessation support with interventions outside the health care system to facilitate tobacco cessation among patients in this setting. Increased tobacco price via tax has been found the most effective intervention in reducing tobacco-related inequalities [41] but increased accessibility and affordability of cessation services through subsidies and decreased availability of tobacco products may further facilitate tobacco cessation among socioeconomically disadvantaged groups [42].

Although the fidelity to the TCP components was relatively high among the informants and they reported that they recognized the content on the TCP prescription form, they were uncertain as to whether they had received the prescription form or not. The informants were invited to participate after they had responded to the 12 month follow-up in the RCT and interviewed up to 19 months after receiving the intervention, indicating that recall bias may have been introduced. To avoid this, future studies could collect data on patients' experiences of TCP closer in time to receiving the intervention. This was not possible in this study as it could have affected the informants' responses to the follow-up questionnaires in the RCT. While it was recommended that the PHC providers in the RCT filled out the TCP prescription form together with the patients and gave them a copy, it was up to the PHC providers to decide how they wanted to use the TCP prescription form. Documentation protocols from the RCT suggest that TCP prescription forms were filled out for all but one of the patients that were interviewed in this study but the authors cannot be certain that the forms were shared with the informants. If they were and the informants forgot about it, this suggests that the TCP prescription form itself did not have a direct impact on tobacco cessation from the patients' perspective. This is consistent with the findings from a study on patients' experiences of PAP in Sweden, showing that the PAP prescription form was forgotten by patients and mainly had a symbolic value compared to the subsequent support provided [43]. Previous studies have shown that advice may be taken more seriously when combined with a prescription form [20, 43] but this was not reported by the informants in this study. However, the informants perceived the TCP prescription form as useful for PHC providers. Similar findings have been reported in another study where the TCP prescription form was also perceived to have an impact on PHC provider behavior with positive outcomes for patients [20]. If TCP is found effective, it may be relevant to implement to facilitate tobacco cessation in the given setting. In this case, it is important that the purpose of the TCP prescription form is clearly communicated to patients as the informants expressed more negative feelings about the prescription form when they did not understand its purpose. However, the informants also engaged with other components of TCP as they received tobacco cessation counseling, options for further support and follow-up. These aspects of TCP were perceived positively, suggesting that the counseling and follow-up components of TCP may be more important to patients than the prescription form itself.

A challenge in this study was that four of the interviews were conducted via telephone, making it more difficult to ask about the TCP prescription form. Three of the four informants that were interviewed via telephone were sent the TCP prescription form in advance. However, it was not possible for the interviewer to ascertain that it was that particular form that the informants were reflecting on without visual confirmation. Still, offering telephone interviews as an alternative to face-to-face interviews enabled several informants to participate that otherwise would have been excluded from the study [44]. Including them was considered a strength since their experiences of tobacco cessation differed from those of the informants that were able to participate in the face-to-face interviews.

The results in this study are consistent with previous studies on socioeconomically disadvantaged smokers' experiences of tobacco cessation in other contexts, suggesting that these

experiences are not specific to Sweden. Therefore, they may also be transferable to cigarette smoking in similar contexts. However, experiences of other tobacco users may be different [36]. For example, snus use may be less stigmatizing [36, 45] as it is not perceived as harmful as smoking [46] and does not expose others to second-hand smoke [45]. Perceived tobacco-related stigma may also be higher in Sweden compared to other settings where the prevalence of tobacco use is higher and more normalized.

The authors' lack of personal experience of tobacco cessation hindered them to personally relate to the experiences that the informants reported. At the same time, this meant that the authors had no preconceived ideas about what the informants would report based their own experiences. Additional differences between the informants and the researchers in characteristics like age, gender, socioeconomic position, language and cultural background, may have introduced misunderstandings or misinterpretations in the data collection and analysis [47]. The power imbalance between the interviewer and the informants and the sensitivity of the topic may also have introduced social desirability bias in the reporting [48]. However, the informants were unaware of the interviewer's role in the development of TCP. Credibility was also enhanced by independent coding of the data by a researcher not involved in the development of TCP, member checking with two representatives of the target group, and reporting according to best practice guidelines for qualitative research [27].

## Conclusions

A holistic approach may be needed in cessation treatment, combined with interventions outside the health care system, to facilitate tobacco cessation among patients in socioeconomically disadvantaged areas in Swedish PHC. The TCP prescription form may be helpful for PHC providers but counseling and follow-up appear to be the most important components of TCP for patients in this setting.

## Supporting information

**S1 Appendix. Interview guide.**
(DOCX)

**S2 Appendix. Prescription form.**
(DOCX)

## Acknowledgments

The authors thank the informants for their valuable contributions to this study, Elizabeth Blum for reviewing the translation of the quotes, the interview guide and the content in Table 2, and Professors Carl Johan Sundberg and Peter Lindgren for their comments on this manuscript.

## Author Contributions

**Conceptualization:** Anne Leppänen, Tanja Tomson.

**Data curation:** Anne Leppänen.

**Formal analysis:** Anne Leppänen, Solvig Ekblad, Tanja Tomson.

**Funding acquisition:** Anne Leppänen, Tanja Tomson.

**Investigation:** Anne Leppänen, Tanja Tomson.

**Methodology:** Anne Leppänen, Solvig Ekblad, Tanja Tomson.

**Project administration:** Anne Leppänen, Tanja Tomson.

**Resources:** Tanja Tomson.

**Supervision:** Solvig Ekblad, Tanja Tomson.

**Validation:** Anne Leppänen, Solvig Ekblad, Tanja Tomson.

**Writing – original draft:** Anne Leppänen.

**Writing – review & editing:** Anne Leppänen, Solvig Ekblad, Tanja Tomson.

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
