## [Decision Letter · Decision Letter 0]

10 Jul 2020

PONE-D-20-05329

Patients’ experiences of tobacco cessation including a prescription approach in Swedish primary health care with a focus on socioeconomically disadvantaged areas

PLOS ONE

Dear Dr. Leppänen,

Thank you for submitting your manuscript to PLOS ONE. After careful consideration, we feel that it has merit but does not fully meet PLOS ONE’s publication criteria as it currently stands. Therefore, we invite you to submit a revised version of the manuscript that addresses the points raised during the review process.

We look forward to receiving your revised manuscript.

Kind regards,

Bronwyn Myers

Academic Editor

PLOS ONE

Journal Requirements:

2. Please ensure you have included the registration number for the clinical trial referenced in the manuscript.

Reviewers' comments:

Reviewer's Responses to Questions

**Comments to the Author**

1. Is the manuscript technically sound, and do the data support the conclusions?

Reviewer #1: Yes

Reviewer #2: Yes

2. Has the statistical analysis been performed appropriately and rigorously? 

Reviewer #1: N/A

Reviewer #2: N/A

3. Have the authors made all data underlying the findings in their manuscript fully available?

Reviewer #1: Yes

Reviewer #2: No

4. Is the manuscript presented in an intelligible fashion and written in standard English?

Reviewer #1: Yes

Reviewer #2: Yes

5. Review Comments to the Author

Reviewer #1: The manuscript, “Patients’ experiences of tobacco cessation including a prescription approach in Swedish primary health care with a focus on socioeconomically disadvantaged areas” uses clear and consistent writing to discuss important topics about how to improve the delivery of tobacco treatment. The discussion focuses on factors related to the implementation of a new instrument to facilitate the delivery of smoking cessation treatment. As smoking cessation treatment is mostly provided by Primary Health Care (PHC) worldwide, perceptions of instrument could enrich the knowledge-base of how to effectively engage the PHC in tobacco treatment in countries beyond Sweden.

Introduction:

The introduction is clear and addresses relevant and updated references. The only suggestion I have is to describe a little bit more about the context in which the Tobacco Cessation on Prescription (TCP) form was implemented. I understood that the form has been tested as part of a clinical trial. However, I’m not sure if the form is also part of the smoking cessation treatment guidelines in Stockholm. If so, did the providers receive training on smoking cessation treatment as well? Is the treatment subsidized?

Methods:

General comments

It would be helpful to understand the results if the authors include more socio-demographic and treatment information from each participant. Consider including this in a table. For instance, how was the engagement of each participant with TCP? Did they just fill out the form or did they receive smoking cessation treatment (medication, counseling, follow up calls, referral to Quitline)? Besides smoking status, consider including other smoking related information, such as type of tobacco used, dependence level and also socioeconomic disadvantage. My understanding was that not all of the 8 participants were in a situation of socioeconomical disadvantage.

Study information and setting

1. It is not clear how participants were selected. Could you explain in a few sentences the characteristics that were assessed to determine participants with best knowledge of the phenomena? Were they purposely from different PHC centers? Could you provide more information on the theory that guides the information power?

Data collection and analysis

1. Describe the authors training and experience with qualitative research instead of mentioning just “some experience”.

2. “Passages in the text related to the aim of the study were extracted from the transcripts”. Was this step also done by 2 researchers and then compared?

3. Describe the number or percentage of coding that were random checked. Also, describe if the authors made changes in the coding after the categories were checked by participants.

Results/discussion:

The way the authors presented the themes and subcategories made the reading smooth and clear. However, I’d suggest presenting the results related to patient’s perceptions of the TCP and in the support received from the PHC center first. The title and introduction get attention for these aspects and I felt like the result section was missing these categories until towards to theme 2. Last, the discussion is supported by the main findings. But, the patients’ perceptions probably differ according to their engagement in the smoking cessation support prescribed in their form. How could this be related to the results found in this study?

Minor comments: In the results, instead of using terms such as “several informants” present the number of participants.

Reviewer #2: This manuscript addresses a topic that is highly relevant to public health interventions- tobacco cessation. A few areas require modification:

1. Please define socio-economic disadvantage clearly

2. A little more information about the trial and the training that the interventionists received will help us interpret the results

3. Methods- these need to follow COREQ guidelines- there are some aspects that are missing. We need to know more about the criteria used to purposefully select participants, more information about the interviewers, and there is no information about reflexivity- among other aspects. Was software used to aid analyses

4. We need more information about the participants and their smoking and quit attempt histories- did any differences emerge based on smoking involvement or other factors?

5.The TCP form was poorly remembered. If the trial is about using this prescription, surely more of the results should focus on what patients remember, what kind of prescription they received and whether they attempted to follow through on it. I am not clear on why they thought it was more useful for providers? Also it raises questions about the implementation of this intervention- this could all be reflected upon more in the discussion.

6. PLOS authors have the option to publish the peer review history of their article (what does this mean?). If published, this will include your full peer review and any attached files.

Reviewer #1: **Yes: **Erica Cruvinel

Reviewer #2: No

---

## [Author Response · Author response to Decision Letter 0]

21 Aug 2020

Authors’ response to reviews: We are very thankful for all the valuable comments that we have received from the reviewers on this manuscript. Our response point by point is presented below. Revised sections have been highlighted in the manuscript using Track Changes. The page and line references below are based on the “Final: Show Markup” setting in Track Changes. An unmarked version of the revised manuscript is also included in the submission.

Reviewer #1: The manuscript, “Patients’ experiences of tobacco cessation including a prescription approach in Swedish primary health care with a focus on socioeconomically disadvantaged areas” uses clear and consistent writing to discuss important topics about how to improve the delivery of tobacco treatment. The discussion focuses on factors related to the implementation of a new instrument to facilitate the delivery of smoking cessation treatment. As smoking cessation treatment is mostly provided by Primary Health Care (PHC) worldwide, perceptions of instrument could enrich the knowledge-base of how to effectively engage the PHC in tobacco treatment in countries beyond Sweden.

Introduction:

The introduction is clear and addresses relevant and updated references. The only suggestion I have is to describe a little bit more about the context in which the Tobacco Cessation on Prescription (TCP) form was implemented. I understood that the form has been tested as part of a clinical trial. However, I’m not sure if the form is also part of the smoking cessation treatment guidelines in Stockholm. If so, did the providers receive training on smoking cessation treatment as well? Is the treatment subsidized? 

Answer: Several studies on TCP have been conducted in primary healthcare (PHC) in Stockholm but it has not been formally implemented in this setting. This has now been clarified in the introduction (page 6, lines 104-105) and methods section (page 7, lines 131-133). Further details on the RCT and its implementation are provided in the trial registration [ISRCTN 11498135] and the published study protocol. This has now been added to the methods section (page 7, lines 128-130). 

Methods:

General comments

It would be helpful to understand the results if the authors include more socio-demographic and treatment information from each participant. Consider including this in a table. For instance, how was the engagement of each participant with TCP? Did they just fill out the form or did they receive smoking cessation treatment (medication, counseling, follow up calls, referral to Quitline)? Besides smoking status, consider including other smoking related information, such as type of tobacco used, dependence level and also socioeconomic disadvantage. My understanding was that not all of the 8 participants were in a situation of socioeconomical disadvantage. 

Answer: Due to the small sample size, the specificity of the population and the time frame, it is not possible to provide such detailed information on each informant without compromising the confidentiality of the informants. However, a table with this information presented on the group level has now been added (pages 8-9, lines 170-172). The aim of this study was to explore experiences of tobacco cessation and TCP among patients in the given setting. Further details on the treatment provided to the patients in the RCT will be reported in a separate manuscript. 

It is correct that not all informants were in a situation of socioeconomic disadvantage at the individual level. Therefore, the aim of the study focused on patients in socioeconomically disadvantages areas (rather than on individuals with socioeconomic disadvantage) (page 6, lines 113-115). 

Study information and setting

1. It is not clear how participants were selected. Could you explain in a few sentences the characteristics that were assessed to determine participants with best knowledge of the phenomena? Were they purposely from different PHC centers? Could you provide more information on the theory that guides the information power? 

Answer: The informants were purposefully sampled based on their willingness to participate in the current study as well as their experience of tobacco cessation and TCP. Since TCP has not been formally implemented in clinical practice, the participants in the RCT, evaluating the effectiveness of TCP in the PHC setting with a focus on socioeconomically disadvantaged areas, were the only patients with experience of this intervention. These patients were also expected to have some experience of trying to quit their tobacco use from their participation in the RCT. Thus, they were considered the most suitable informants for the current study. The procedure for sampling and recruitment of informants has now been clarified in the methods section (page 7, lines 124-141). 

Patients were invited from five of the intervention PHC centers and recruited from three intervention PHC centers in the RCT. This has now been added to the methods section (page 8, lines 160-162). This was the result rather than the intention of the sampling strategy.

Due to the inductive nature of the current study, it was not relevant to apply any theory to guide the data analysis. This has now been clarified (page 8, lines 155-156). The concept of information power has been developed based on available literature about the current state of the art regarding sample size in qualitative studies and weaknesses in these standards. This has now been added to the methods section (page 8, lines 151-153). 

Data collection and analysis

1. Describe the authors training and experience with qualitative research instead of mentioning just “some experience”. 

Answer: AL has formal training in qualitative research methods and previous experience from two qualitative studies, SE has extensive experience in teaching and conducting qualitative research on vulnerable groups in different cultural settings and TT has experience from several qualitative studies in different contexts globally. This has now been added to the methods section (pages 10-11, lines 189-192, 206-210). 

2. “Passages in the text related to the aim of the study were extracted from the transcripts”. Was this step also done by 2 researchers and then compared? 

Answer: Yes, this has now been clarified (page 11, lines 211-212). 

3. Describe the number or percentage of coding that were random checked. Also, describe if the authors made changes in the coding after the categories were checked by participants. 

Answer: Approximately one third of the codes were compared. No changes were made to the coding as a result of the member checking. This has now been added to the methods section (page 11, lines 217-223). 

Results/discussion:

The way the authors presented the themes and subcategories made the reading smooth and clear. However, I’d suggest presenting the results related to patient’s perceptions of the TCP and in the support received from the PHC center first. The title and introduction get attention for these aspects and I felt like the result section was missing these categories until towards to theme 2. Last, the discussion is supported by the main findings. But, the patients’ perceptions probably differ according to their engagement in the smoking cessation support prescribed in their form. How could this be related to the results found in this study? 

Answer: The authors agree that it would be beneficial to report the results related to TCP and tobacco cessation in PHC first. However, the results related to TCP are included in a sub-category in the first theme and the results related to tobacco cessation in PHC are included in a sub-category in the second theme so it is not possible to report both of these sub-categories in the beginning of the results. Since these sub-categories are logically related to each other, the authors argue that these should be reported in close proximity to each other. To enable this and the logical flow of the results, the authors argue that the current order of the sub-categories and themes should be kept as is (TCP in the last sub-category in Theme 1 and tobacco cessation in PHC in the first sub-category in Theme 2). 

Regarding the suggestion for the discussion, it may be the case that patients’ perceptions differ according to their engagement in the smoking cessation support prescribed in their form. However, this was not mentioned by the informants during the interviews. As stated in the discussion, the informants mainly reflected on their general experiences of trying to quit their tobacco use, i.e. not only the cessation support they had received in the RCT but also their previous experiences of trying to quit with support from the health care system and otherwise. The title and aim of the study has now been revised to better reflect that the study explored the experiences of tobacco cessation and TCP among patients in the given setting and not only the cessation support received in the RCT and/or in PHC (page 1 and 6, lines 113-115). 

Minor comments: In the results, instead of using terms such as “several informants” present the number of participants. 

Answer: Since this is a qualitative study and the aim of this type of research is to describe the variety and patterns of experiences among the informants and not to quantify them, it is considered more appropriate to use this type of general terms in the reporting of the results than writing the exact number of informants that have described a particular experience. Therefore, the authors argue that the current language in the results should be kept as is. 

Reviewer #2: This manuscript addresses a topic that is highly relevant to public health interventions- tobacco cessation. A few areas require modification: 

1. Please define socio-economic disadvantage clearly 

Answer: In this study, socioeconomically disadvantaged groups are defined as groups with a lower socioeconomic status, e.g. lower income, educational level or occupational status. This has now been added to the introduction (page 4, lines 57-61). 

2. A little more information about the trial and the training that the interventionists received will help us interpret the results 

Answer: Since the focus of this study was on experiences of tobacco cessation and TCP among patients in the given setting and not on the RCT or the PHC providers delivering the intervention, the authors refer to the trial registration and published study protocol for further details on the RCT and its implementation. This has now been added to the methods section (page 7, lines 128-130). 

3. Methods- these need to follow COREQ guidelines- there are some aspects that are missing. We need to know more about the criteria used to purposefully select participants, more information about the interviewers, and there is no information about reflexivity- among other aspects. Was software used to aid analyses 

Answer: The informants were purposefully sampled based on their willingness to participate in the current study as well as their experience of tobacco cessation and TCP. Since TCP has not been formally implemented in clinical practice, the participants in the RCT, evaluating the effectiveness of TCP in the PHC setting with a focus on socioeconomically disadvantaged areas, were the only patients with experience of this intervention. These patients were also expected to have some experience of trying to quit their tobacco use from their participation in the RCT. Thus, they were considered the most suitable informants for the current study. The procedure for sampling and recruitment of informants has now been clarified in the methods section (page 7, lines 124-141). Further details about the authors and the use of software to aid the data analysis have also been added (pages 10-11, lines 189-192, 206-210, 222-223). Moreover, aspects of reflexivity have now been expanded in the discussion (pages 26-27, lines 562-569). 

4. We need more information about the participants and their smoking and quit attempt histories- did any differences emerge based on smoking involvement or other factors? 

Answer: Further details on the informants are now presented at the group level in Table 1 (pages 8-9, lines 170-172). The aim of this study was to explore experiences of tobacco cessation and TCP among patients in the given setting. The patterns that were identified from a qualitative perspective have already been presented in the results and discussion. Patterns will also be analyzed from a quantitative perspective in a separate manuscript regarding the RCT. 

5. The TCP form was poorly remembered. If the trial is about using this prescription, surely more of the results should focus on what patients remember, what kind of prescription they received and whether they attempted to follow through on it. I am not clear on why they thought it was more useful for providers? Also it raises questions about the implementation of this intervention- this could all be reflected upon more in the discussion. 

Answer: The results related to TCP have now been reported in further detail (pages 17-18, lines 330-336). However, it is discussed in the manuscript that the informants mainly reflected on their general experiences of trying to quit their tobacco use, i.e. not only the cessation support they had received in the RCT but also their previous experiences of trying to quit with support from the health care system and otherwise. The title and aim of the study has now been revised to better reflect that the study explored the experiences of tobacco cessation and TCP among patients in the given setting and not only the cessation support received in the RCT and/or in PHC (page 1 and 6, lines 113-115). 

Information about the fidelity to the TCP components has been added in Table 1 and the discussion (page 8-9, lines 170-172 and page 24, lines 510-513). Methodological considerations related to the limited reporting on TCP among the informants (risk of recall bias, etc.) and the implications of this are also discussed on pages 24-25, lines 510-541. 

Journal Requirements 

Answer: The manuscript has now been updated to meet PLOS ONE's style requirements, including those for file naming. 

2. Please ensure you have included the registration number for the clinical trial referenced in the manuscript. 

Answer: A reference to the trial registration for the RCT has now been added to the methods section (page 7, lines 128-130). 

Answer: The data set cannot be shared publicly due to ethical and legal restrictions since it contains potentially identifying and sensitive information that compromises the informants' confidentiality. This would violate the ethical approval obtained from the Regional Ethical Review Board in Stockholm [ref: 2015/207-31] as well as the data protection policy of Karolinska Institutet. Data are available on request from the Registrar's Office at Karolinska Institutet (contact via registrator@ki.se) for researchers who meet the criteria for access to confidential data.

---

## [Decision Letter · Decision Letter 1]

28 Sep 2020

Experiences of tobacco cessation including a prescription approach among patients in Swedish primary health care with a focus on socioeconomically disadvantaged areas

PONE-D-20-05329R1

Dear Dr. Leppänen,

We’re pleased to inform you that your manuscript has been judged scientifically suitable for publication and will be formally accepted for publication once it meets all outstanding technical requirements.

Kind regards,

Bronwyn Myers

Academic Editor

PLOS ONE

Additional Editor Comments (optional):

Reviewers' comments:

Reviewer's Responses to Questions

**Comments to the Author**

1. If the authors have adequately addressed your comments raised in a previous round of review and you feel that this manuscript is now acceptable for publication, you may indicate that here to bypass the “Comments to the Author” section, enter your conflict of interest statement in the “Confidential to Editor” section, and submit your "Accept" recommendation.

Reviewer #1: All comments have been addressed

Reviewer #2: All comments have been addressed

2. Is the manuscript technically sound, and do the data support the conclusions?

Reviewer #1: Yes

Reviewer #2: Yes

3. Has the statistical analysis been performed appropriately and rigorously? 

Reviewer #1: Yes

Reviewer #2: N/A

4. Have the authors made all data underlying the findings in their manuscript fully available?

Reviewer #1: Yes

Reviewer #2: Yes

5. Is the manuscript presented in an intelligible fashion and written in standard English?

Reviewer #1: Yes

Reviewer #2: Yes

6. Review Comments to the Author

Reviewer #1: It appears that the authors thoroughly addressed the reviewer's concerns. I have no further revisions.

Reviewer #2: I am happy with the revisions that have been made. All original concerns have been fully addressed.

7. PLOS authors have the option to publish the peer review history of their article (what does this mean?). If published, this will include your full peer review and any attached files.

Reviewer #1: No

Reviewer #2: No

---

## [Editor Report · Acceptance letter]

1 Oct 2020

PONE-D-20-05329R1 

Experiences of tobacco cessation including a prescription approach among patients in Swedish primary health care with a focus on socioeconomically disadvantaged areas 

Dear Dr. Leppänen:

I'm pleased to inform you that your manuscript has been deemed suitable for publication in PLOS ONE. Congratulations! Your manuscript is now with our production department. 

Kind regards, 

on behalf of

Dr. Bronwyn Myers 

Academic Editor

PLOS ONE